

# Large-Scale Tropospheric Transport in the Chemistry Climate Model Initiative (CCMI) Simulations

Clara Orbe[1,2,3], Huang Yang[2], Darryn W. Waugh[2], Guang Zeng[4], Olaf Morgenstern [4], Douglas E. Kinnison[5], Jean-Francois Lamarque[5], Simone Tilmes[5], David A. Plummer[6], John F. Scinocca[7], Beatrice Josse[8], Virginie Marecal[8], Patrick Jöckel[9], Luke D. Oman[10], Susan E. Strahan[10,11], Makoto Deushi[12], Taichu Y. Tanaka[12], Kohei Yoshida[12], Hideharu Akiyoshi[13], Yousuke Yamashita[13,14], Andreas Stenke[15], Laura Revell[15,16], Timofei Sukhodolov[15,17], Eugene Rozanov[15,17], Giovanni Pitari[18], Daniele Visioni[18], Kane A. Stone[19,20,21], and Robyn Schofield[19,20]

[1]GESTAR
[2]Department of Earth and Planetary Sciences, Johns Hopkins University, Baltimore, Maryland, USA
[3]Global Modeling and Assimilation Office, NASA Goddard Space Flight Center, Greenbelt, Maryland, USA
[4]National Institute of Water and Atmospheric Research, Wellington, New Zealand
[5]National Center for Atmospheric Research (NCAR), Atmospheric Chemistry Observations and Modeling (ACOM) Laboratory, Boulder, USA
[6]Climate Research Branch, Environment and Climate Change Canada, Montreal, QC, Canada
[7]Climate Research Branch, Environment and Climate Change Canada, Victoria, BC, Canada
[8]Centre National de Recherches Météorologiques UMR 3589, Météo-France/CNRS, Toulouse, France
[9]Deutsches Zentrum für Luft- und Raumfahrt (DLR), Institut für Physik der Atmosphäre, Oberpfaffenhofen, Germany
[10]Atmospheric Chemistry and Dynamics Laboratory, NASA Goddard Space Flight Center, Greenbelt, Maryland, USA
[11]Universities Space Research Association
[12]Meteorological Research Institute (MRI), Tsukuba, Japan
[13]Climate Modeling and Analysis Section, Center for Global Environmental Research, National Institute for Environmental Studies, Tsukuba, Japan
[14]Japan Agency for Marine-Earth Science and Technology (JAMSTEC), Yokohama, Japan
[15]Institute for Atmospheric and Climate Science, ETH Zürich (ETHZ), Switzerland
[16]Bodeker Scientific, Christchurch, New Zealand
[17]Physikalisch-Meteorologisches Observatorium Davos / World Radiation Centre, Davos, Switzerland
[18]Department of Physical and Chemical Sciences, Universitá dell'Aquila, Italy
[19]School of Earth Sciences, University of Melbourne, Melbourne, Victoria 3010, Australia
[20]ARC Centre of Excellence for Climate System Science, University of New South Wales, Sydney, New South Wales 2052, Australia
[21]Now at the Department of Earth, Atmospheric and Planetary Sciences, Massachusetts Institute of Technology, Cambridge, Massachusetts 02139-4307, USA.

*Correspondence to:* Clara Orbe (clara.orbe@nasa.gov)

**Abstract.** Understanding and modeling the large-scale transport of trace gases and aerosols is important for interpreting past (and projecting future) changes in atmospheric composition. Here we show that there are large differences in the global-scale atmospheric transport properties among models participating in the IGAC SPARC Chemistry-Climate Model Initiative (CCMI). Specifically, we find up to 40% differences in the transport timescales connecting the Northern Hemisphere (NH) 5 midlatitude surface to the Arctic and to Southern Hemisphere high latitudes, where the mean age ranges between 1.7 years



and 2.6 years. We show that these differences are related to large differences in vertical transport among the simulations and, in particular, to differences in parameterized convection over the oceans. While stronger convection over NH midlatitudes is associated with slower transport to the Arctic, stronger convection in the tropics and subtropics is associated with faster interhemispheric transport. We also show that the differences among simulations constrained with fields derived from the same reanalysis products are as large as (and, in some cases, larger than) the differences among free-running simulations, due to larger differences in parameterized convection. Our results indicate that care must be taken when using simulations constrained with analyzed winds to interpret the influence of meteorology on tropospheric composition.

# 1 Introduction

The distributions of greenhouse gases (GHG) and ozone-depleting substances (ODS) are strongly influenced by large-scale atmospheric transport. In the extratropics the midlatitude jet stream influences the long-range transport of pollutants and water vapor into the Arctic (e.g., *Eckhardt et al.*, 2003; *Shindell et al.*, 2008; *Liu and Barnes*, 2015), as well as surface ozone variability over the Western United States (*Lin et al.*, 2015). In the tropics, low-level inflow and seasonal variations in the Hadley Cell modulate trace gas variability in the tropics and interhemispheric transport into the Southern Hemisphere (SH) (*Prather et al.*, 1987; *Mahlman*, 1997; *Holzer*, 1999; *Bowman and Erukhimova*, 2004).

There are large uncertainties in our understanding of how large-scale atmospheric transport influences tropospheric composition. This is largely because transport is difficult to constrain directly from observations and because global-scale tropospheric transport properties differ widely among models. For example, *Denning et al.* (1999) found more than a factor of two difference in the interhemispheric exchange rate among simulations produced using both offline chemical transport models (CTMs) and online free-running general circulation models (GCMs).

One approach to reducing this uncertainty has been to use models constrained with analysis fields, although comparisons of the transport properties among these simulations also reveal large differences. For example, *Patra et al.* (2011) showed that the interhemispheric transport differences among CTMs participating in the TransCOM experiment differ by up to a factor of two, with models featuring faster interhemispheric transport also exhibiting faster exchange of methane and methyl chloroform. It is not clear, however, whether these differences reflect subgrid-scale differences among CTMs or differences in the prescribed large-scale flow, since that study included simulations that were constrained with three different sources of meteorological fields.

More recently, *Orbe et al.* (2017) compared the global-scale tropospheric transport properties among free-running simulations using internally generated meteorological fields as well as simulations constrained with analysis fields using models developed at NASA Goddard Space Flight Center and the Community Earth System Model framework (CESM) (run at the National Center for Atmospheric Research (NCAR)). They showed that the large-scale transport differences among simulations constrained with analysis fields are as large as (and, in some cases, larger than) the differences among free-running simulations. Furthermore, they found that these differences – manifest over southern high latitudes as a 0.6 year (or ∼ 30%) difference in the mean age since air was last at the Northern Hemisphere (NH) midlatitude surface – were associated with large differences in



(parameterized) convection, particularly over the NH tropics and subtropics. By comparison, the mean age differences between the free-running simulations were found to be negligible, consistent with much more similar convective mass fluxes.

The results in *Orbe et al.* (2017) indicate that care must be taken when using simulations constrained with analysis fields to interpret the influence of meteorology on tropospheric composition. It is not clear, however, if the conclusions from that study reflect only that particular subset of models and/or the particular ways in which those models were constrained with analysis fields. To this end we exploit the broad range of both free-running online and offline (i.e. nudged and CTM) simulations submitted to the recent IGAC/SPARC Chemistry-Climate Model Initiative (CCMI) (*Eyring et al.*, 2013) in order to test some of the key findings in that study. In particular, we focus on the CCMI hindcast simulations of the recent past, which include simulations constrained with both prescribed and internally generated meteorological fields, while sea surface temperatures (SSTs) and sea ice concentrations (SICs) are taken from observations. Thus, the CCMI hindcast experiment provides a relatively clean framework for assessing the influence of different meteorological fields on large-scale atmospheric transport.

As in *Orbe et al.* (2017) we focus on large-scale tropospheric transport diagnosed from idealized tracers that, unlike the usual basic flow diagnostics (e.g. mean winds, streamfunctions, mean eddy diffusivities), represent the integrated effects of advection and diffusion while cleanly disentangling the roles of transport from chemistry and emissions. Furthermore, unlike previous intercomparisons that have diagnosed atmospheric transport in terms of one single timescale (e.g. the interhemispheric exchange rate (*Denning et al.*, 1999; *Patra et al.*, 2011)), we utilize tracers with different prescribed atmospheric lifetimes and different source regions in order to probe the broad range of timescales and pathways over which tropospheric transport occurs (*Orbe et al.*, 2016). Following a brief exposition of the methodology in Section 2 we present results in Sections 3 and conclusions in Section 4.

## 2 Methods

### 2.1 Models and Experiments

Our analysis uses models participating in CCMI, which builds upon previous chemistry-climate model intercomparisons, including the SPARC Report on the Evaluation of Chemistry-Climate Models (*CCMVal*, 2010) and The Atmospheric Chemistry and Climate Model Intercomparison Project (ACCMIP) (*Lamarque et al.*, 2013), by including several coupled atmosphere-ocean models with a fully resolved stratosphere. For example, more (nine) models are atmosphere-ocean (versus only one in CCMVal-2 and one in ACCMIP) and more models incorporate novel (e.g. cubed-sphere) grids (*Morgenstern et al.*, 2017).

We focus only on those CCMI model simulations that output the idealized tracers (Table 1, Table 2). We present results from the pair of hindcast REF-C1 (simply C1) and REF-C1SD (or C1SD) simulations, which were constrained with observed SSTs and SICs. Whereas the REF-C1 experiment simulates the recent past (1960-2010) using internally generated meteorological fields, the REF-C1SD or C1 "Specified Dynamics" simulation is constrained with (re)analysis meteorological fields and, correspondingly, only spans the years 1980-2010. Note that both online nudged simulations as well as offline CTMs are used, as indicated in the simulation name. Furthermore, while we have also examined tracer output from the REF-C2 simulation, which used SSTs from a coupled atmosphere-ocean model simulation, we find that the differences in the idealized tracers between




the REF-C2 and REF-C1 simulations are significantly smaller than among the hindcast (C1 versus C1SD) simulations. For that reason, hereon we exclude the REF-C2 results from our discussions.

The simulations presented in *Orbe et al.* (2017) using models from NASA and NCAR are included in our analysis and denoted in all figures using a color convention that is similar to what was used in that study. Note that this subset of runs includes two REF-C1SD simulations per modeling group. In particular, the GEOS-CTM and GEOS-C1SD simulations refer to one simulation of the NASA Global Modeling Initiative (GMI) Chemical Transport Model (*Strahan et al.*, 2013) and one simulation of the Goddard Earth Observing System General Circulation Model Version 5 (GEOS-5) (*Reinecker et al.*, 2007; *Molod et al.*, 2015), respectively; they are both constrained with fields taken from The Modern-Era Retrospective Analysis for Research and Applications (MERRA) (*Rienecker et al.*, 2011). Meanwhile, the WACCM-C1SDV1 and WACCM-C1SDV2 correspond to two simulations of the Whole Atmosphere Community Climate Model (*Marsh et al.*, 2013) nudged to MERRA meteorological fields using two different relaxation timescales (i.e. 50 hours and 5 hours).

In addition to differences among the REF-C1 and REF-C1SD experiments, the models differ widely in terms of their horizontal resolution, which ranges from $\sim 6$ degrees (e.g. ULAQ) to $\sim 2$ degrees (e.g. NCAR and NASA), vertical resolution, and choices of sub-grid scale (i.e. turbulence and convective) parameterizations (*Morgenstern et al.*, 2017). Table 1 summarizes some of the main differences among the models, as well as the method by which the large-scale flow was constrained in the REF-C1SD simulations (i.e. CTM versus nudging). For more details please refer to the comprehensive overview presented in *Morgenstern et al.* (2017).

Finally, we complement our analysis of the idealized tracers with comparisons of the models' convective mass fluxes, horizontal and vertical winds, and temperature fields (when available) (Table 3). All tracer and dynamical variables were available as monthly mean output on native model levels. Therefore, we interpolated all output to a standard pressure vector with 4 pressure levels in the stratosphere (10, 30, 50 and 80 hPa) and 19 pressure levels in the troposphere spaced every 50 hPa between 100 hPa and 1000 hPa. To construct all of the multi-model means (denoted in the figures using solid grey lines) we first interpolated all model output to the same one-degree latitude by one-degree longitude grid and then took the average among the models. As in *Orbe et al.* (2017) our focus is on seasonal averages over December-January-February (DJF) and June-July-August (JJA) and on ten-year climatological means over the time period 2000-2009, which are denoted throughout using overbars.

## 2.2 Idealized Tracers

Several of the idealized tracers examined in this study (Table 2) were discussed in *Orbe et al.* (2016, 2017). Figure 1 shows boreal winter (DJF) and boreal summer (JJA) climatological mean distributions of the tracers for one model simulation, which has been chosen purely for illustrative purposes. This is the GEOS-CTM simulation that was presented in *Orbe et al.* (2017) and described in the previous section. Schematic representations of the seasonally averaged mean meridional circulation, overlaid with arrows denoting mixing by eddies, are also shown to help guide the interpretation of the tracer distributions (Fig. 1f).

Three of the tracers' boundary conditions are zonally uniform and are defined over the same NH surface region over midlatitudes, $\Omega_{\mathrm{MID}}$, which we define as the first model level spanning all grid points between $30°N$ and $50°N$ (rows 2-4 in Table 2,



Figure 1 a-b). The first two tracers, $\chi_5$ and $\chi_{50}$, referred to throughout as the 5-day and 50-day idealized loss tracers, are fixed to a value of 100 ppb over $\Omega_{\mathrm{MID}}$ and undergo spatially uniform exponential loss at rates of 5 days$^{-1}$ and 50 days$^{-1}$, respectively. The climatological mean distributions of the loss tracers, denoted throughout as $\overline{\chi}_5$ and $\overline{\chi}_{50}$, decrease poleward away from the midlatitude source region during boreal winter, when tracer isopleths coincide approximately with isentropes that intersect the Earth's surface, reflecting the strong influence of isentropic mixing on surface source tracer distributions over middle and high latitudes (Fig 1f). During summer, by comparison, the idealized loss tracer patterns extend higher into the upper troposphere over midlatitudes, consistent both with weaker isentropic transport over the northern extratropics and stronger convection over the continents (*Klonecki et al.*, 2003; *Stohl*, 2006; *Orbe et al.*, 2015). Compared to $\overline{\chi}_5$, which is mainly confined to the NH extratropics, large values of $\overline{\chi}_{50}$ span the NH subtropics and tropics.

The third NH midlatitude tracer, $\Gamma_{\mathrm{NH}}$, is initially set to a value of zero throughout the troposphere and held to zero thereafter over $\Omega_{\mathrm{MID}}$ (Fig.1c). Elsewhere over the rest of the model surface layer and throughout the atmosphere $\Gamma_{\mathrm{NH}}$ is subject to a constant aging of 1 year/year so that its statistically stationary value, the mean age, is equal to the average time since the air at a given location in the troposphere last contacted the NH midlatitude surface $\Omega_{\mathrm{MID}}$ (*Waugh et al.*, 2013). The strongest meridional gradients in the mean age $\Gamma_{\mathrm{NH}}$, which increases from $\sim 3$ months in the NH extratropical lower troposphere to $\sim 2$ years over SH high latitudes, are located in the tropics and migrate north and south in concert with seasonal shifts in the Intertropical Convergence Zone (ITCZ) and the mean meridional circulation (Fig. 1f) (*Waugh et al.*, 2013).

In addition to the NH midlatitude source tracers, we also examine two other tracers with global sources. The first tracer, $\chi_{\mathrm{STE}}$, is set to a constant value of 200 ppb above 80 hPa and undergoes spatially uniform exponential loss at a rate of 25 days$^{-1}$ in the troposphere. The second tracer, e90 is uniformly emitted over the surface layer and decays exponentially at a rate of 90 days $^{-1}$ such that mixing ratios greater than 125 ppb tend to reside in the lower troposphere and mixing ratios smaller than 50 ppb reside in the stratosphere (*Prather et al.*, 2011). While their mean gradients are opposite in sign, due to differences in their boundary conditions, both tracers feature pronounced signatures of isentropic transport in the subtropical upper troposphere along isentropes spanning the middleworld (*Hoskins*, 1991). This is evident in the plume of large mixing ratios of $\chi_{\mathrm{STE}}$ and, conversely, small concentrations of e90, that extends down from the tropopause to the subtropical surface (Fig. 1 d-e). The seasonality of this isentropic transport is captured by the relatively larger (smaller) values of $\chi_{\mathrm{STE}}$ (e90) in the northern subtropical upper troposphere during winter, compared to during summer (and vice versa in the SH).

# 3 Results

## 3.1 Transport to Northern Hemisphere High Latitudes

### 3.1.1 Differences in Transport

Meridional profiles of $\overline{\chi}_5$ and $\overline{\chi}_{50}$, averaged over the middle troposphere (400-700 mb), differ widely among the simulations over the NH extratropics (Figure 2). Over northern midlatitudes $\overline{\chi}_5$ differs by up to a factor of 5 during boreal winter and a factor of 2-3 during boreal summer. The spread in the 50-day loss tracer, $\overline{\chi}_{50}$, is similar, consistent with the strong compact



relationship between the loss tracers, such that simulations featuring low concentrations of $\overline{\chi}_5$ also feature low concentrations of $\overline{\chi}_{50}$ (and vice versa) (see also Figure 4a below). During summer, the differences in $\overline{\chi}_{50}$ extend all the way to the pole, where $\overline{\chi}_{50}^{\mathrm{JJA}}$ ranges between ~20-50 ppb among the simulations. Note that these differences are overall much larger than the differences among the simulations presented in *Orbe et al.* (2017) (red and blue lines, Fig. 2), which feature consistently larger

concentrations of $\overline{\chi}_5$ and $\overline{\chi}_{50}$ over northern middle and high latitudes compared to the other simulations, indicative of more efficient poleward transport in those models.

Interestingly, the differences in the concentrations of $\overline{\chi}_5$ and $\overline{\chi}_{50}$ among the C1SD simulations are as large as the differences among the C1 simulations. For example, the 5-day loss tracer concentrations over midlatitudes range between 9-22 ppb during boreal summer among both the C1 and C1SD simulations (Fig. 2b). During boreal winter the spread among the C1 simulations

is slightly larger, but closer inspection shows that this only reflects the inclusion of one outlier simulation (Fig. 2a,c). Overall, this is consistent with *Orbe et al.* (2017), who found that the transport differences between two simulations of GEOS-5 and WACCM constrained with fields taken from MERRA were as large as (and, at places, larger than) the differences between free-running simulations generated using the same models.

Comparisons of the global source tracer e90 also reveal large differences among the simulations (Figure 3). The spread in e90

mixing ratios is similar in magnitude to the spread in the concentrations of the idealized loss tracers, especially over northern midlatitudes, where there is an inverse (and relatively compact) relationship among the tracers (Figure 4b). This is consistent with differences in isentropic transport such that simulations with relatively large mixing ratios of $\chi_5$ and $\chi_{50}$ (blue and red lines in Figure 2) also feature relatively small mixing ratios of e90 (Figure 3 a-b) due to stronger dilution of surface (high e90) concentrations with upper tropospheric (low e90) air masses. By comparison, simulations featuring slower isentropic transport

(i.e. ULAQ C1), are characterized by relatively smaller (larger) values of $\chi_5$ (e90).

The spread in $\chi_{\mathrm{STE}}$ among the CCMI simulations is also large (Figure 3c-d). However, care must be taken when interpreting differences in $\chi_{\mathrm{STE}}$ as solely reflecting differences in stratosphere-troposphere-exchange. In particular, the distribution of $\chi_{\mathrm{STE}}$ in the outlier simulations (i.e. NIWA C1 and ACCESS C1) may reflect the fact that these models use a hybrid-height vertical coordinate such that the tracer's 80 hPa upper boundary condition is not parallel to any model level and, therefore, more

easily communicated to lower levels (Supplementary Figure 1). Furthermore, while the NIWA and ACCESS simulations use essentially the same model, we note that there are small differences between them that may reflect differences in computing platforms.

Among the other simulations, by comparison, the differences in $\chi_{\mathrm{STE}}$ emerge below the tropopause, where they more likely reflect differences in isentropic mixing in the subtropical upper troposphere. Among those simulations there is a relatively

compact relationship between $\overline{\chi}_{\mathrm{STE}}^{\mathrm{DJF}}$ and $\overline{\mathrm{e}90}^{\mathrm{DJF}}$ during boreal winter over the northern subtropical upper troposphere (Figure 4c), consistent with the *Abalos et al.* (2017) analysis of the WACCM-SD simulation. Similar to the findings in that study, our results suggest that both tracers may be useful metrics for discerning stratosphere-troposphere-exchange differences among models. Finally, comparisons of the spatial distributions of $\chi_{\mathrm{STE}}$ fail to reveal any consistencies with differences in tropopause height among the simulations, which are not negligible (Supplementary Figure 2). This indicates that differences in tropopause

height are not likely to be the primary drivers of the $\chi_{\mathrm{STE}}$ differences within the CCMI ensemble.





Zonal profiles of $\overline{\chi}_5$ reveal that differences in the loss tracer distributions over the Arctic reflect differences in isentropic transport originating over the northern subtropical oceans (Figure 5). During winter large differences in $\overline{\chi_5}^{\text{DJF}}$ emerge over the oceans in the lower troposphere (900 mb) (Fig. 5a) and propagate along isentropes towards high latitudes downstream of the stormtracks (Fig. 5e). By comparison, during boreal summer, the large differences in $\overline{\chi}_5^{\text{JJA}}$ that emerge in the subtropics over

land (Fig. 5b) remain relatively confined over midlatitudes. Rather, the differences in $\overline{\chi}_5^{\text{JJA}}$ over the Arctic, more likely reflect differences that emerge over the midlatitude oceans over the northern edge of the source region (Fig. 5d). We interpret these transport differences next in terms of differences in the large-scale flow and (parameterized) convection among the simulations.

### 3.1.2   Differences in Northern Midlatitude Convection and Large-Scale Flow

One approach to interpreting the large differences in poleward transport among the CCMI simulations is to compare the (pa-

rameterized) convection and horizontal flow fields over northern midlatitudes (Figure 6). During winter the multi-model mean convective mass fluxes ($\overline{\text{CMF}}^{\text{DJF}}$) in the lower troposphere (700-900 mb) (Fig. 6a) are concentrated over the Pacific and Western Atlantic (black boxes). These regions coincide with the climatological mean position of warm conveyer belts at the midlatitude jet entrance regions (*Eckhardt et al.*, 2004) as well as with low values of potential temperature ($280K{<}\theta{<}290K$) approximately along which surface mixing ratios of $\overline{\chi_5}^{\text{DJF}}$ propagate poleward into the upper and middle high latitude tropo-

sphere (Fig.1a; Fig. 6b).

By comparison, during boreal summer the (parameterized) convective mass fluxes are generally weaker over midlatitudes and shift from the oceans toward land, coincident with weaker and zonally shifted storm tracks. Seasonal changes in the thermal structure of the extratropics also indicate that the Arctic is isentropically isolated from the northern midlatitude surface during summer, compared to winter (*Klonecki et al.*, 2003). The CCMI simulations capture this seasonality well, in terms of both the

convective mass flux distributions (Fig. 6c) and in the redistribution of potential temperature surfaces (Fig. 6d). Any differences in transport among the simulations that emerge over the northern midlatitude surface, therefore, are more likely to be confined to the midlatitude upper troposphere during boreal summer, compared to during winter.

Comparisons of vertical profiles of the convective mass fluxes (CMF) over northern midlatitudes (black boxed regions in Figure 6), reveal large differences in (parameterized) convection among the models during both boreal winter and summer

(Figure 7). Among the "weak midlatitude convection" simulations (i.e. NASA and NCAR), the strength of $\overline{\text{CMF}}^{\text{DJF}}$ is at places half (West Pacific) and one third (West Atlantic) the strength in the "strong midlatitude convection" simulations (i.e. NIWA, ACCESS and EMAC). Note that the latter simulations use convection parameterizations that have a diagnostic closure scheme based on large-scale convergence (i.e. based on that of *Tiedtke* (1989)), whereas the former simulations' utilize relaxed and/or triggered adjustment schemes in which adjustments to explicitly defined moist-convective equilibrium states are partly

relaxed (*Arakawa*, 2004) (Table 1). While the former class of parameterizations tends to produce excessive precipitation relative to observations (*Garcia et al.*, 2017) further analysis of the differences among the models' convection schemes is beyond the scope of this study.

Closer inspection of the loss tracer profiles at 30°N during boreal winter reveals that simulations with strong convection over the oceans also feature steeper vertical profiles of $\overline{\chi}_5$, compared to models with weaker convection (not shown). This reflects the





influence of convective updrafts mixing large near-surface concentrations aloft and convective downdrafts mixing low upper-tropospheric concentrations to the surface (*Zhang et al.*, 2008). As a result, among simulations with stronger (parameterized) convective mass fluxes we find overall *smaller* concentrations of $\overline{\chi_5}^{\mathrm{DJF}}$ at the midlatitude surface and, correspondingly, smaller concentrations over the Arctic, compared to simulations with weaker convection over the midlatitude oceans.

This is illustrated more clearly in Figure 8, which shows strong negative correlations during boreal winter between lower tropospheric (800-950 mb) convection ($\overline{\mathrm{CMF}}^{\mathrm{DJF}}$), evaluated over the midlatitude oceans, and zonal mean concentrations of $\overline{\chi_5}^{\mathrm{DJF}}$, averaged poleward of 60°N and over the middle troposphere (Figure 8 a-c). The strong negative correlations indicate that models with weak convection over the oceans are associated with more efficient transport to the Arctic (i.e. less surface dilution and larger mixing ratios of $\overline{\chi_5}^{\mathrm{DJF}}$). Note that this relationship is robust among the CCMI simulations over various
ocean basins despite (large) interannual variability over the 2000-2009 climatological period examined in this study.

We also find evidence of a relationship between midlatitude convection and the loss tracer concentrations over the Arctic during boreal summer, although this relationship is relatively weaker (Figure 8 d-f). This most likely reflects the fact that the Arctic is isentropically isolated from the northern midlatitude surface during boreal summer, compared to during winter (*Klonecki et al.*, 2003). Preliminary analyses indicate that differences in the northern boundary of the Hadley Cell among the
simulations may also play an important role in understanding the differences in poleward transport during boreal summer, as discussed further in *Yang et al. (2017, In Prep)*. In contrast, comparisons of the pressure velocity $\omega$ among the models do not reveal a consistent relationship between large-scale flow biases over NH midlatitudes and the transport differences among the simulations for either season (Supplementary Figure 3).

### 3.2    Interhemispheric Transport

### 3.2.1    Differences in Transport

We now compare different measures of interhemispheric transport among the models. As in *Orbe et al.* (2016) we recast the idealized loss tracer concentrations $\chi_5$ and $\chi_{50}$ in terms of "tracer ages" $\tau_5$ and $\tau_{50}$, where $\tau_T(\mathbf{r}, t) = -T\ln(\frac{\chi_{\mathrm{T}}(\mathbf{r},\mathrm{t})}{\chi_\Omega})$, $\Omega$ is the NH midlatitude source region $\Omega_{\mathrm{MID}}$ and T refers to the exponential decay timescales 5 days and 50 days, respectively. This is a common approach in oceanography and facilitates comparison with the NH midlatitude mean age $\Gamma_{\mathrm{NH}}$ (*Deleersnijder et al.*,
2001; *Waugh and Hall*, 2002).

Meridional profiles of the annually averaged $\tau_5$, $\tau_{50}$ and $\Gamma_{\mathrm{NH}}$ reveal large differences among all of the tracer ages over the middle troposphere (300-600 mb) (Figure 9), with Southern Hemisphere (SH) values of $\tau_5$ ranging between 70 days and 90 days, or $\sim 25\%$ of the multimodel mean, while the mean age $\Gamma_{\mathrm{NH}}$ varies between 1.7 years and 2.6 years, or about $\sim 40\%$ of the multimodel mean. The differences in the tracer ages among the simulations emerge primarily in the tropics and are
more-or-less consistent among the different ages such that simulations that tend to have small values of $\tau_5$ (relative to the multi-model mean) also feature relatively small values of the mean age $\Gamma_{\mathrm{NH}}$. This indicates that the age tracer differences arise due to transport differences in the tropical and subtropical lower troposphere and not in response to differences in the lower stratosphere, to which the 5-day age tracer is insensitive.





Consistent with the results in *Orbe et al.* (2017) we find that the interhemispheric transport differences among the C1SD simulations are as large as the differences among the free-running C1 simulations. Interestingly this applies not only to simulations constrained with MERRA analysis fields (i.e. GEOS-CTM, GEOS-C1SD, WACCM C1SDV1/V2, and CAM C1) but also simulations constrained with fields from ERA-Interim (i.e. CMAM-C1SD, MOCAGE-CTM and NIES-C1SD). For example, the mean age differs by $\sim 0.5$ years between the MOCAGE-CTM and CMAM-C1SD simulations over the SH, compared to only about 0.15 years between the GEOS-C1 and CMAM-C1 free-running simulations, despite substantial differences in the large-scale flow among those models.

While $\Gamma_{NH}$ cannot be observed directly, *Waugh et al.* (2013) show that it can be approximated in terms of the time lag between the mixing ratio of sulfur hexaflouride ($SF_6$) at a given location and the NH midlatitude surface. We have confirmed this finding among three of the CCMI simulations, for which both $SF_6$ and $\Gamma_{NH}$ were output (not shown). Furthermore, comparisons with observational estimates of $\Gamma_{NH}$, inferred in *Waugh et al.* (2013) from surface measurements of $SF_6$, indicate that all of the CCMI models are old relative to the observations. The paucity of $SF_6$ output among the models, however, precludes a more detailed model-observation comparison in this study.

### 3.2.2 Differences in Tropical Large-Scale Flow and Parameterized Convection

A possible source of differences in interhemispheric transport among the C1SD simulations are differences in the analysis fields themselves, which can differ significantly among reanalysis products (*Stachnik and Schumacher*, 2011). A comparison of the large-scale flow in the tropics, however, reveals much larger differences among the C1 simulations, where we have approximated the tropical meridional circulation in terms of the meridional and vertical components of the velocity field (Figure 10). This applies both to comparisons of the upper tropospheric meridional flow (V) (Fig. 10 a-d) as well as comparisons of the pressure velocity ($\omega$) among the simulations, although the differences in $\omega$ among the C1SD simulations are by no means negligible (Fig. 10 e-h). Furthermore, the differences among the NCAR and NASA C1SD simulations are small, despite the fact that the differences in the mean age $\Gamma_{NH}$ among those simulations spans most of the ensemble spread (Figure 9). Overall this suggests that the interhemispheric transport differences among the simulations are not driven to first order by differences in the large-scale flow.

Rather, *Orbe et al.* (2017) show that differences in interhemispheric transport between the NASA and NCAR C1SD simulations are related to differences in convection over the northern subtropical oceans. We test this result among all the CCMI models and expand our region of interest to also include latitudes in the deep tropics, in accordance with previous studies showing that deep tropical convection significantly enhances interhemispheric transport (*Gilliland and Hartley*, 1998).

Among the CCMI ensemble we find strong correlations between anually averaged lower tropospheric (700-900 mb) convection over the tropical oceans and Southern Hemisphere tracer ages averaged poleward of $60°$S (Figure 11). Consistent with large differences in interhemispheric transport among the C1SD simulations, Figure 11 reveals large differences in parameterized convection among simulations constrained with analysis fields. Furthermore, note that, while the correlations are shown for the annual mean, we have performed a similar analysis accounting for seasonal variations in convection. That analysis reveals similar (if stronger) correlations (not shown).



## 4 Conclusions

Comparisons of idealized tracers among the CCMI hindcast simulations reveal large differences in their global-scale tropospheric transport properties. In particular:

- There are large (30-40%) differences in the efficiency of transport from the Northern Hemisphere midlatitude surface into the Arctic. To first order, these differences reflect differences in (parameterized) convection over the northern midlatitude oceans, particularly during boreal winter.

- There are large differences in interhemispheric transport from northern midlatitudes to southern high latitudes, where the mean age $\Gamma_{NH}$ ranges between 1.7 years and 2.6 years. In general, stronger tropical and subtropical convection is associated with faster interhemispheric transport.

- The large-scale transport differences among simulations constrained with analyzed winds are as large as the differences among simulations using internally generated meteorological fields, consistent with the findings in *Orbe et al.* (2017). This is because the differences in (parameterized) convection among specified-dynamics simulations can be larger than the differences among free-running simulations.

Our findings suggest that differences in parameterized convection over the oceans are the primary drivers of transport differences among the CCMI simulations. By comparison, we show that the differences related to how the large-scale flow is specified (e.g. CTM vs. nudging or source of analysis fields) are relatively smaller. Therefore, our results indicate that caution should be taken when using the C1SD simulations to interpret the influence of meteorology on tropospheric composition. In the future more attention will need to be paid to understanding the behavior of convective parameterizations in simulations constrained with analyzed winds, both in offline (CTM) and online (nudged) frameworks.

At this point it is not clear why the convection differences among the C1SD simulations are in certain cases larger than among free-running simulations using the same models. One possibility is that these differences arise due to inconsistencies (e.g. in resolution or unbalanced dynamics) between the driving large-scale flow fields and the convective mass fluxes, which are recalculated online in all of the nudged simulations as well as in the MOCAGE-CTM, or interpolated directly from analysis fields (e.g. GEOS-CTM). The analysis in this study has been limited by the small number of C1SD simulations that output all of the idealized tracers as well as convective mass fluxes (Table 3). Experiments using multiple sources of analysis fields as well as different convective parameterizations will need to be performed in order to examine this problem more carefully. A review of the CCMI C1SD simulations, with details of how these simulations were constrained, is also currently in preparation and may provide further insight.

One important caveat in this study is that our focus has been on tracers with zonally uniform boundary conditions. The implications of our findings will, therefore, vary among different species, depending on where they are emitted over the Earth's surface. In particular, our results highlight the differences in transport that arise due to large differences in (parameterized) oceanic convection among the simulations. We anticipate, therefore, that our results will primarily apply to species with oceanic





sources, including marine-sourced volatile organic compounds and short-lived ozone-depleting halogenated species. By comparison, species with primarily land emissions (e.g. short-lived species) are expected to be more sensitive to other aspects of transport. To this end, a study is currently in preparation which addresses the implications of biases in the latitude of the midlatitude jet on carbon monoxide distributions over the Arctic among the CCMI models. We reserve further discussions to

that study.

*Acknowledgements.* We thank the Centre for Environmental Data Analysis (CEDA) for hosting the CCMI data archive. We acknowledge the modeling groups for making their simulations available for this analysis, and the joint WCRP SPARC/IGAC Chemistry-Climate Model Initiative (CCMI) for organizing and coordinating this model data analysis activity. In addition, C.O. and L.D.O. want to thank the high-performance computing resources provided by the NASA Center for Climate Simulation (NCCS) as well as support from the NASA Mod-

eling, Analysis and Prediction (MAP) program. H.A. acknowledges Environment Research and Technology Development Fund of the Environmental Restoration and Conservation Agency, Japan (2-1709) and NECSX9/A(ECO) computers at CGER, NIES. O.M. and G. Z. acknowledge the UK Met Office for use of the MetUM. Their research was supported by the NZ Government's Strategic Science Investment Fund (SSIF) through the NIWA programme CACV. OM acknowledges funding by the New Zealand Royal Society Marsden Fund (grant 12-NIW-006) and by the Deep South National Science Challenge (*http://www.deepsouthchallenge.co.nz*). O.M. and G.Z. also wish

to acknowledge the contribution of NeSI high-performance computing facilities to the results of this research. New Zealand's national facilities are provided by the New Zealand eScience Infrastructure (NeSI) and funded jointly by NeSI's collaborator institutions and through the Ministry of Business, Innovation and Employment's Research Infrastructure programme (*https://www.nesi.org.nz*). D.W. acknowledges support from NSF grant AGS-1403676 and NASA grant NNX14AP58G. The EMAC model simulations have been performed at the German Climate Computing Centre (DKRZ) through support from the Bundesministerium für Bildung und Forschung (BMBF). DKRZ and its scien-

tific steering committee are gratefully acknowledged for providing the HPC and data archiving resources for the consortial project ESCiMo (Earth System Chemistry integrated Modeling). ER and TS acknowledge support from the Swiss National Science Foundation under grant 200021169241 (VEC). RS and KS acknowledge support from Australian Research Council's Centre of Excellence for Climate System Science (CE110001028), the Australian Government's National Computational Merit Allocation Scheme (q90) and Australian Antarctic science grant program (FoRCES 4012).





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



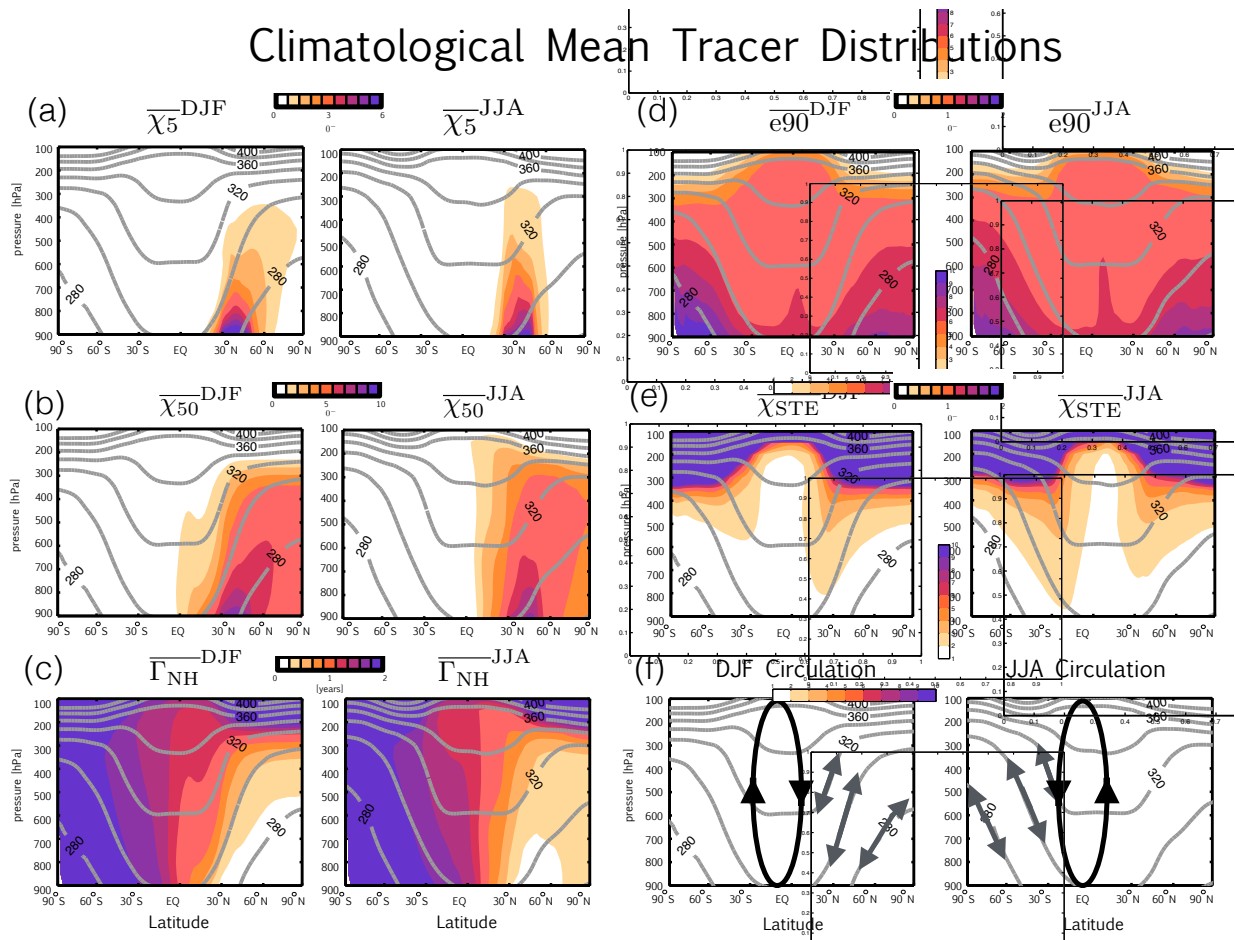

**Figure 1.** Climatological mean December-January-February (DJF) (left panels) and June-July-August (JJA) (right panels) zonally averaged distributions of the 5-day idealized loss tracer, $\chi_5$ (a), the 50-day idealized loss tracer $\chi_{50}$ (b), the mean age since air was last at the NH midlatitude surface $\Gamma_{NH}$ (c), the stratospheric global source tracer $\chi_{STE}$ (d) and the global surface source tracer e90 (e). Schematic representations of the seasonally averaged mean meridional circulation, overlaid with arrows denoting eddy mixing, are shown in panel f. 2000-2009 climatological means are shown for the NASA Global Modeling Initiative (GMI) Chemical Transport Model (CTM), which is constrained with MERRA meteorological fields and denoted in all remaining figures as the GEOS-CTM simulation. Climatological seasonal mean dry potential temperature is shown in the grey contours.





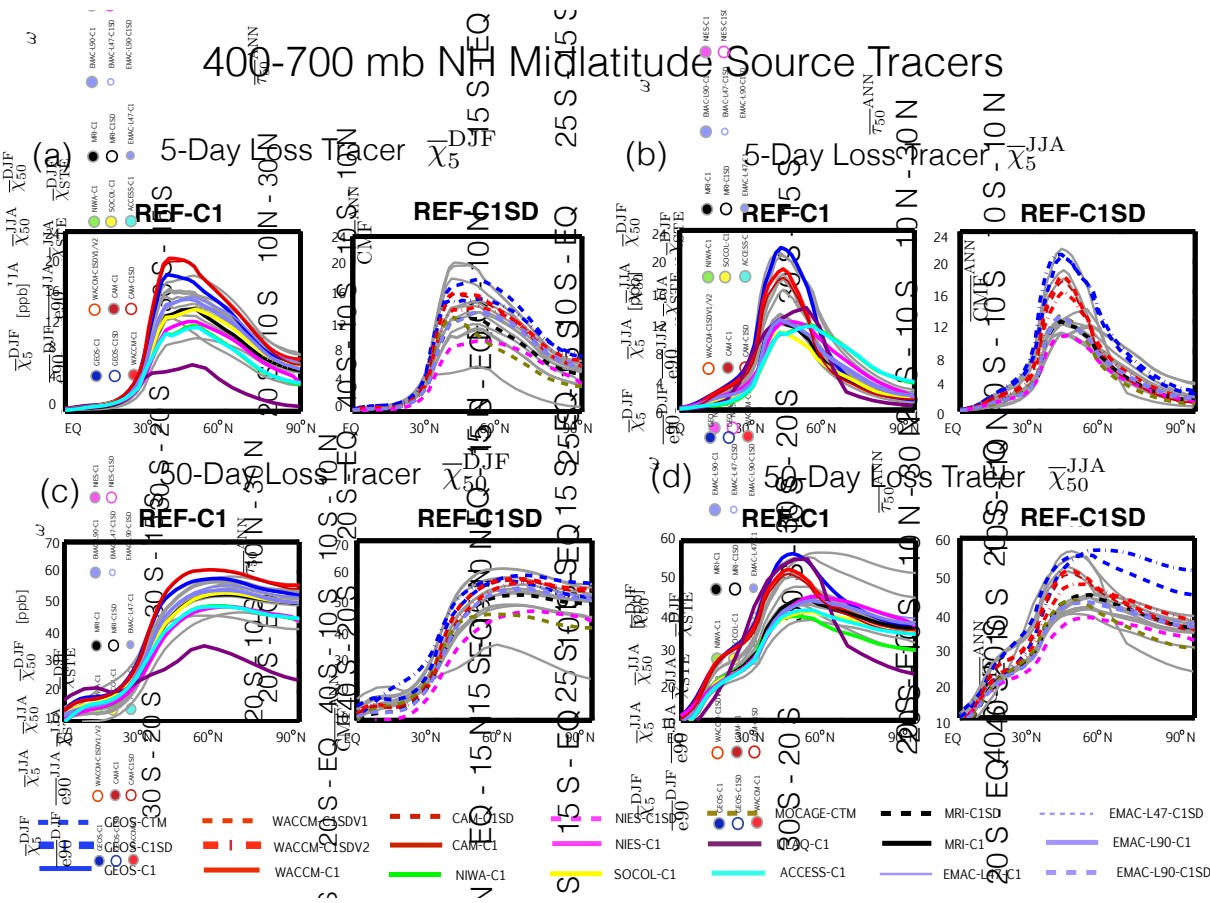

**Figure 2.** Meridional profiles of the 400-700 hPa zonally averaged DJF (a,c) and JJA (b,d) 5-day and 50-day loss tracers, $\overline{\chi}_5$ and $\overline{\chi}_{50}$. Dashed lines in each right panel correspond to the REF-C1SD simulations, which are constrained with analysis meteorological fields, while solid lines in each left panel correspond to the free-running REF-C1 simulations. Grey lines in each panel correspond to the REF-C1SD and REF-C1 simulations in the left and right panels, respectively. Note that the x-axis only spans the Northern Hemisphere.





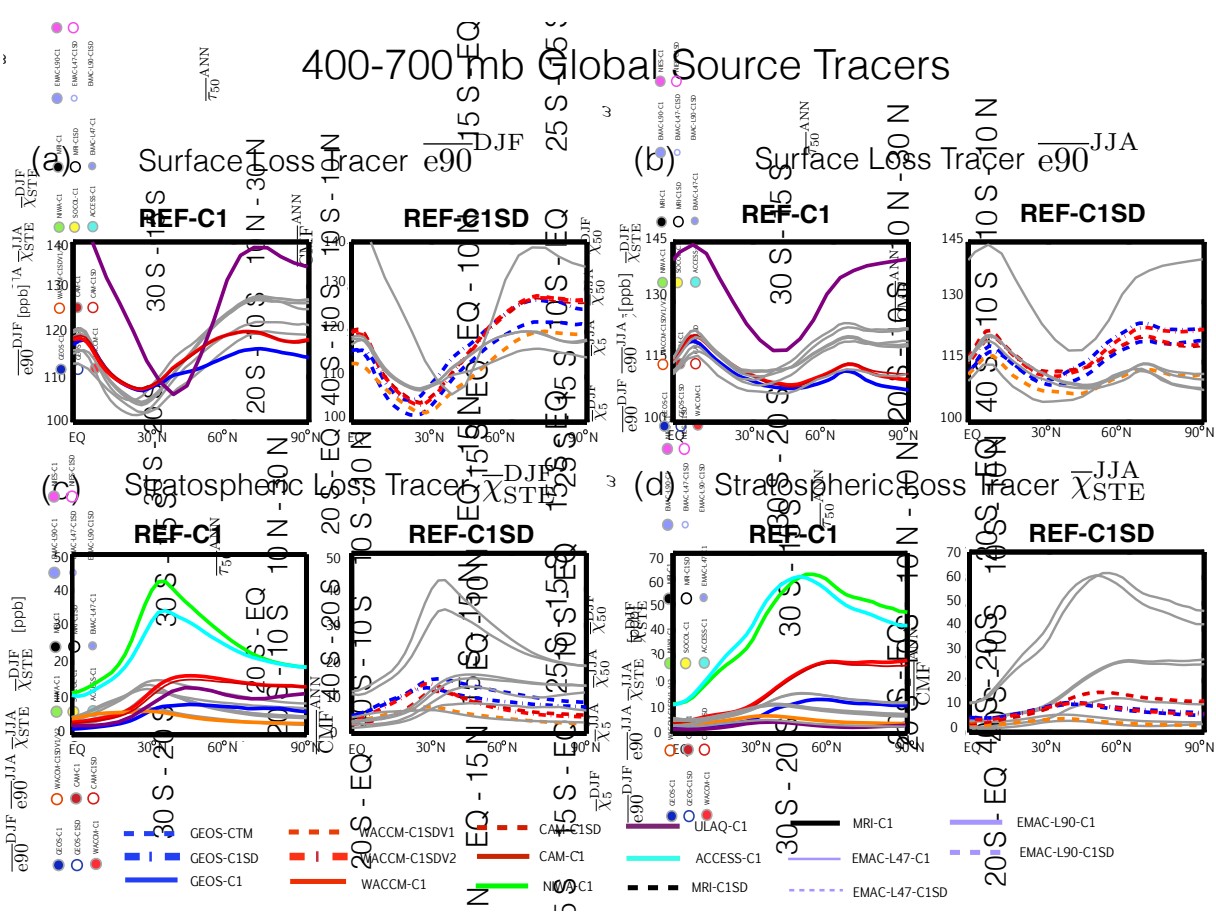

**Figure 3.** Same as Figure 2, except for the stratospheric and surface global source tracers, e90 (top) and $\chi_{\mathrm{STE}}$ (bottom).



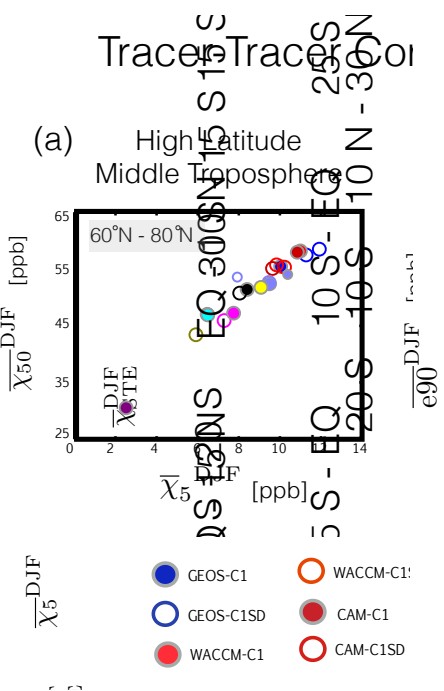

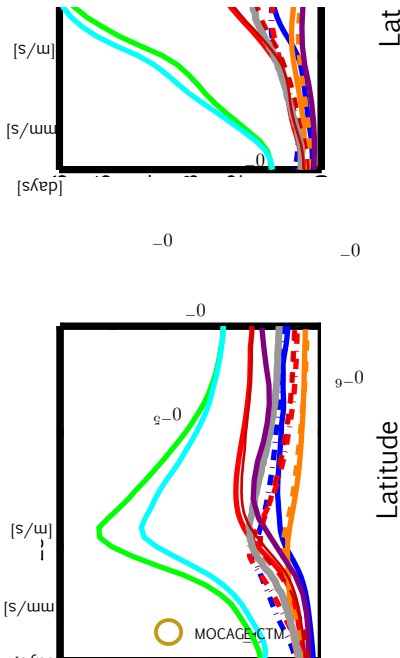

**Figure 4.** (a) Correlations of 400-700 mb averages of $\overline{\chi_5}^{\mathrm{DJF}}$ and $\overline{\chi_{50}}^{\mathrm{DJF}}$ averaged over latitudes spanning 60°N and 80°N. (b) Correlations of $\overline{\chi_5}^{\mathrm{DJF}}$ and $\overline{e90}^{\mathrm{DJF}}$, averaged over 700-900 mb and over the midlatitude source region. (c) Same as (b), except for $\overline{\chi_{\mathrm{STE}}}^{\mathrm{DJF}}$ and $\overline{e90}^{\mathrm{DJF}}$ and over 400-700 mb. The different colors correspond to the different simulations, with open circles denoting REF-C1SD simulations and closed circles corresponding to REF-C1 (grey outline). Circles denote the climatological boreal winter mean over 2000-2009.





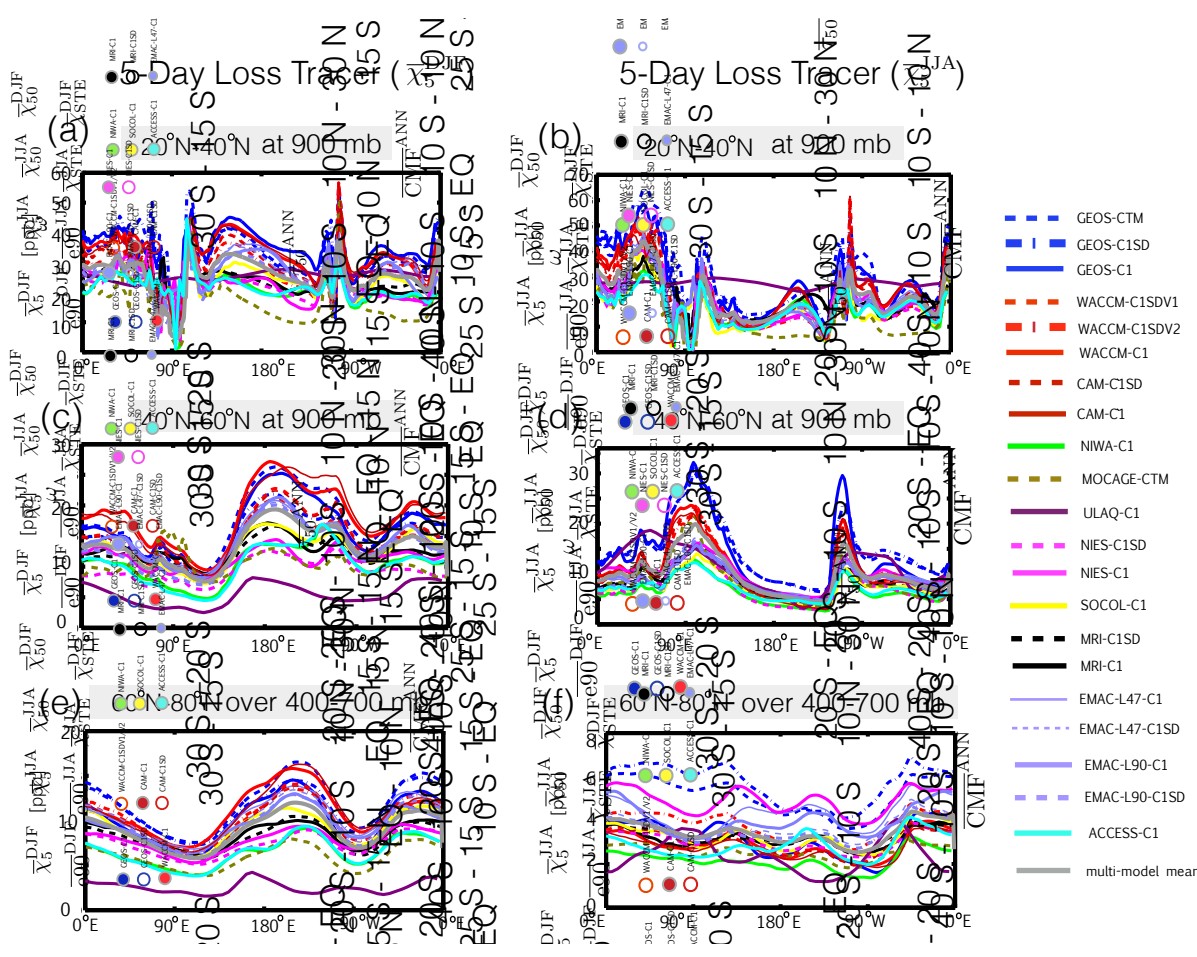

**Figure 5.** Zonal profiles of the climatological mean 5-day idealized loss tracer $\overline{\chi}_5$ averaged over 900 hPa and over latitudes spanning 20°N-40°N (a-b) and 40°N-60°N (c-d) and over 400-700 hPa over latitudes between 60°N-80°N (e-f). Profiles are shown for DJF (left) and JJA (right). Dashed lines correspond to the REF-C1SD simulations, which are constrained with analysis meteorological fields, while solid lines correspond to the free-running REF-C1 simulations. The grey line represents the multimodel mean.





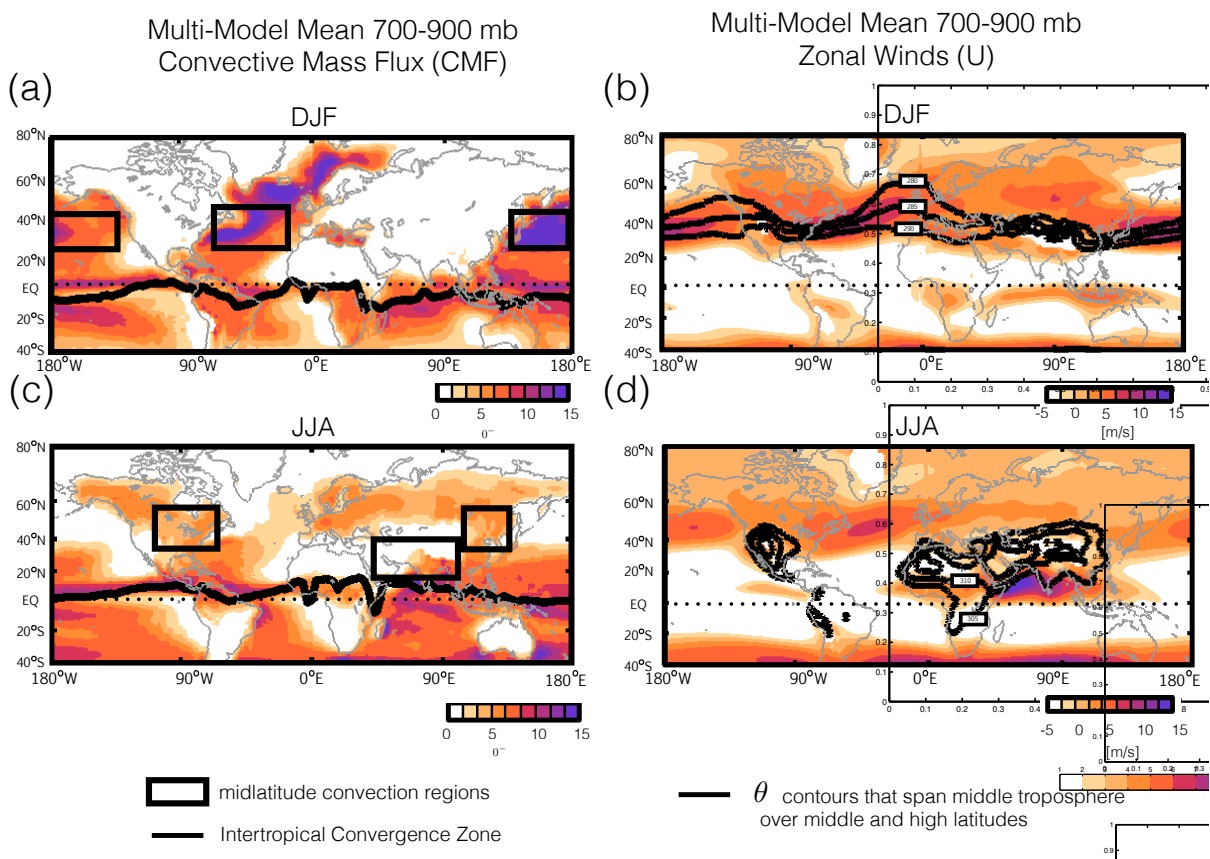

**Figure 6.** Maps of the 700-900 hPa averaged multi-model mean convective mass flux (a,c) and zonal winds (d,e) for DJF (a,b) and JJA (c,d). The multi-model seasonal mean Intertropical Convergence Zone, calculated as the latitude of maximum surface convergence, is shown in the thick black lines in the left panels. Black boxes denote the midlatitude convection regions over which the scatterplots in Figures 8 are evaluated. The thick dark lines in the right panels correspond to the regions where the potential temperature surfaces that span the middle and high latitude upper troposphere intersect the NH midlatitude surface, as shown in Figure 1.



DJF (Top) and JJA (Bottom)
Convective Mass Flux (CMF) Profiles over NH Midlatitudes

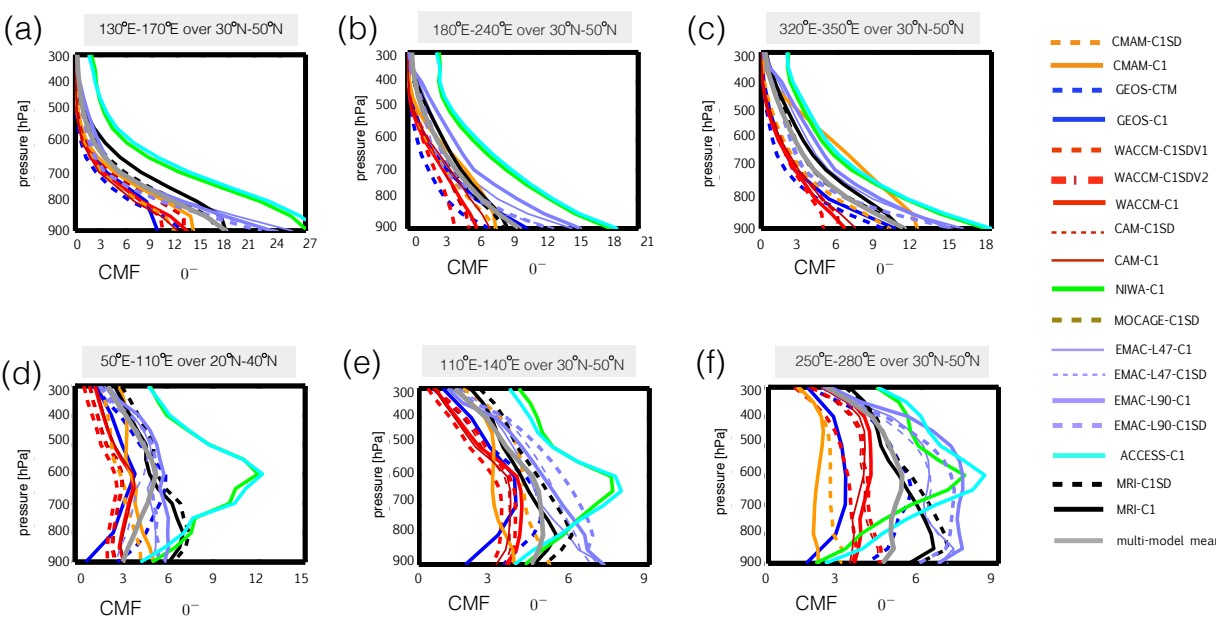

**Figure 7.** Vertical profiles of the convective mass flux evaluated over regions of strong midlatitude convection (black boxes in Figure 6) during DJF (a-c) and JJA (d-f). The thick grey line represents the multimodel mean and dashed lines correspond to the REF-C1SD simulations, while solid lines correspond to the free-running REF-C1 simulations

.





**Figure 8.** Scatterplots showing negative correlations between the strength of parameterized convection in the midlatitude lower troposphere, represented by the 800-950 hPa averaged convective mass flux ($\overline{\text{CMF}}$), and mid-tropospheric (400-700 mb) concentrations of the 5-day idealized loss tracer averaged poleward of 60°N. The convection regions coincide with black boxed regions shown in Figure 6. The different colors correspond to the different simulations, with open circles denoting REF-C1SD simulations and closed circles corresponding to REF-C1 (grey outline) simulations. Small circles correspond to individual years within the 2000-2009 climatological mean period, while large circles denote the climatological mean.



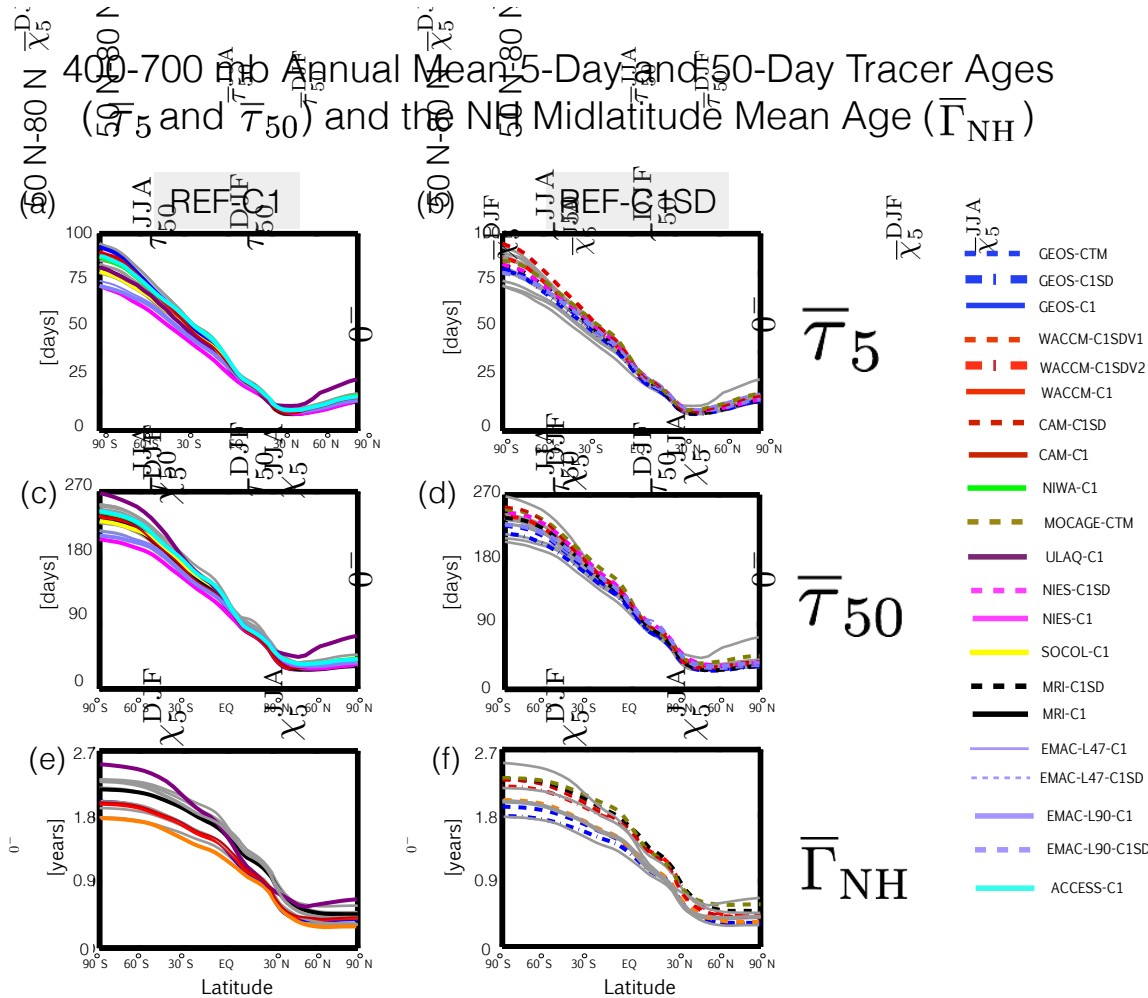

**Figure 9.** Meridional profiles of the annual mean 5-day loss and 50-day loss tracer ages, $\overline{\tau}_5$ (a-b) and $\overline{\tau}_{50}$ (c-d), as well as the annually averaged mean transit time since air was last at the NH midlatitude surface $\overline{\Gamma}_{NH}$ (e-f). Left and right panels show the tracer ages for the REF-C1 and REF-C1SD simulations, respectively. The grey lines denote the C1SD(C1) simulations in the left (right) panels in order to a provide a sense for the ensemble spread.



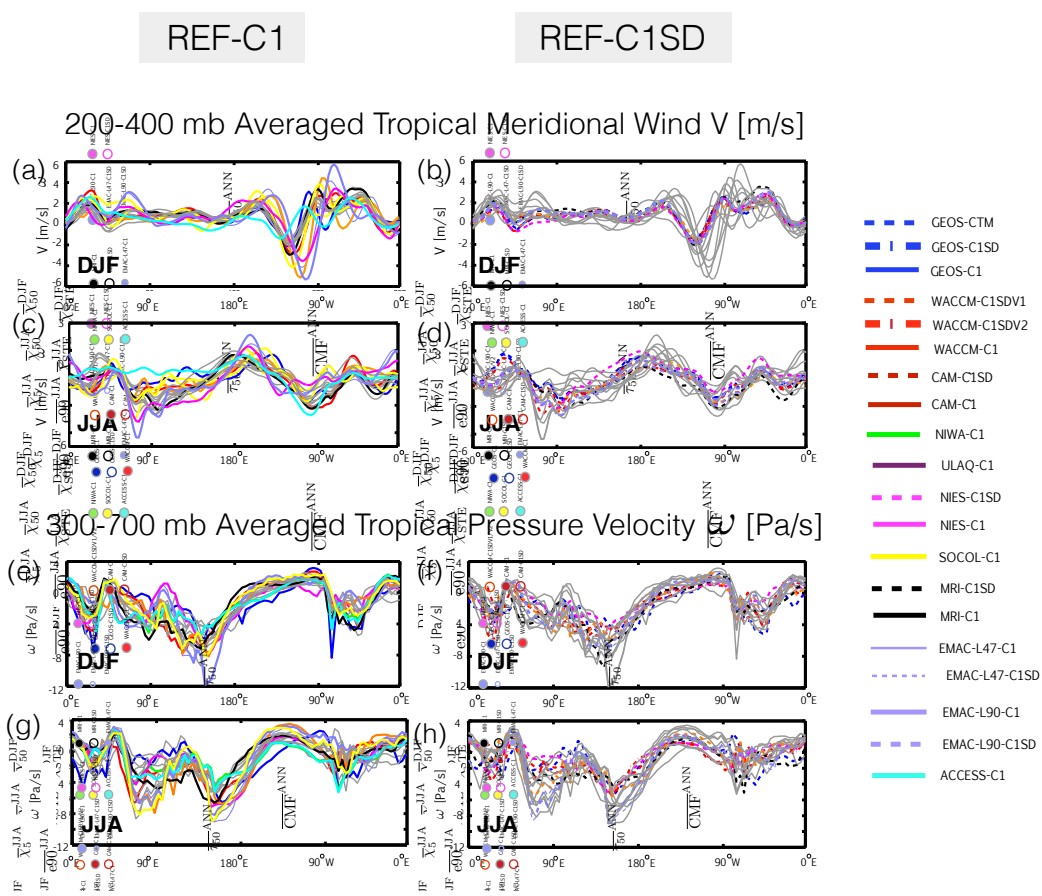

**Figure 10.** Comparisons of the upper tropospheric meridional wind V (a-d) and 300-700 mb averaged pressure velocity (e-h) among the simulations. The REF-C1 and REF-C1SD simulations are shown in the left and right panels, respectively. The grey lines denote the C1SD (C1) simulations in the left (right) panels in order to a provide a sense for the ensemble spread.



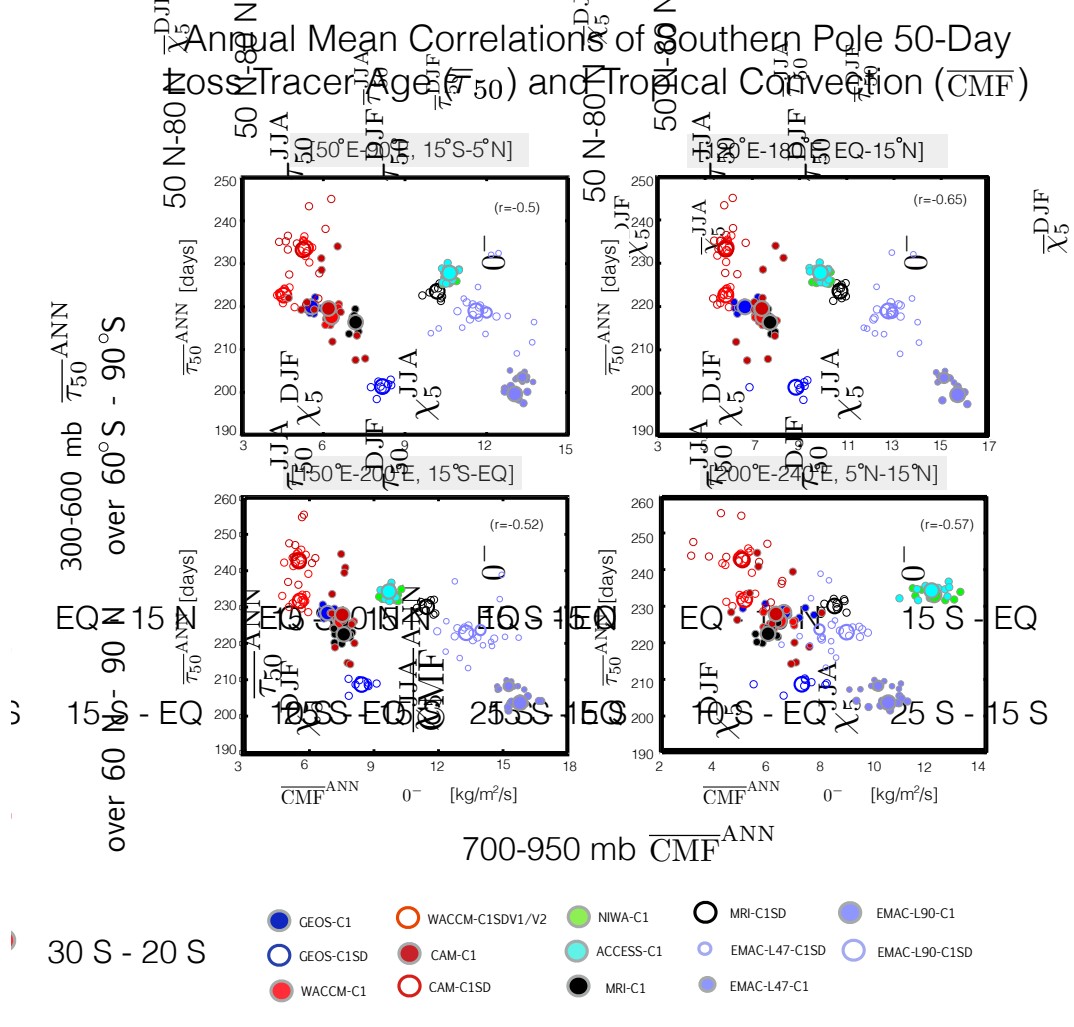

**Figure 11.** Scatterplots showing strong negative correlations between the strength of parameterized convection in the tropics, represented by the 700-900 hPa averaged convective mass flux ($\overline{CMF}$), and mid-tropospheric (300-600 mb) values of the 50-day idealized loss tracer age, $\overline{\tau}_{50}$, evaluated over the Southern Pole. The different colors correspond to the different simulations, with open circles denoting REF-C1SD simulations and closed circles corresponding to REF-C1 free-running simulations. Small circles correspond to individual years within the 2000-2009 climatological mean period, while large circles denote the climatological mean.





# Simulations

| Simulation Name | Model (Reference) | Horizontal Resolution | Vertical Levels (Model Top) | Large-Scale Flow (Free/Nudging/CTM) | Convective Parameterization |
|---|---|---|---|---|---|
| GEOS-CTM | NASA Global Modeling Initiative Chemical Transport Model (Strahan et al., (2013)) | 2° x 2.5° | 72 (0.01 hPa) | MERRA (CTM) | Moorthi and Suarez (1992), Bacmeister et al. (2006) |
| GEOS-C1SD | Goddard Earth Observing System Version 5 GCM (Reinecker et al. (2007), Molod et al. (2015)) | " " | " " | MERRA (Nudging) | " " |
| GEOS-C1 | " " | " " | " " | Free-Running | " " |
| WACCM-C1SDV1 | Whole Atmosphere Community Climate Model Version 4, (WACCM4) (Marsh et al. (2013); Solomon et al. (2015); Garcia et al. (2016)) | 1.9° x 2.5° | 88 (140 km) | MERRA (Nudging) | Hack (1994) (shallow) Zhang and MacFarlane (1995) (deep) |
| WACCM-C1SDV2 | " " | " " | " " | MERRA (Nudging) | " " |
| WACCM-C1 | " " | " " | " " | Free-Running | " " |
| CAM-C1SD | Community Atmosphere Model Version 4 (CAM4)-Chem (Tilmes et al. (2015)) | 1.9° x 2.5° | 56 (1 Pa) | MERRA (Nudging) | " " |
| CAM-C1 | " " | " " | " " | Free-Running | " " |
| EMAC-L47-C1 | ECHAM/ Modular Earth Submodel System (MESSy) Atmospheric Chemistry (EMAC) (Jöckel et al. (2010); Jöckel et al. (2016)) | T42 | 47 (0.01 hPa) | Free-Running | Tiedtke (1989); Nordeng (1994) |
| EMAC-L47-C1SD | " " | " " | " " | ERA-Interim (Nudging) | " " |
| EMAC-L90-C1 | " " | " " | 90 (0.01 hPa) | Free-Running | " " |
| EMAC-L90-C1SD | " " | " " | " " | ERA-Interim (Nudging) | " " |
| MRI-C1SD | Earth System Model MRI-ESM1r1 (Yukimoto et al. (2012, 2011); Deushi and Shibata (2011)) | TL159 | 80 (0.01 hPa) | JRA-55 (Nudging) | Yoshimura et al. (2015) |
| MRI-C1 | " " | " " | " " | Free-Running | " " |
| CMAM-C1SD | Canadian Middle Atmosphere Model (CMAM) (Jonsson et al. (2004); Scinocca et al. (2008)) | T47 | 71 (0.0008 hPa) | ERA-Interim (Nudging) | Zhang and McFarlane (1995) |
| CMAM-C1 | " " | " " | " " | Free-Running | " " |
| NIWA-C1 | National Institute of Water and Atmospheric Research UK Chemistry and Aerosols (NIWA-UKCA) (Morgenstern et al. (2009, 2013); Stone et al. (2016)) | 3.75° x 2.5° | 60 (84 km) | Free-Running | Hewitt et al. (2011) |
| SOCOL-C1 | Solar-Climate-Ozone Links (SOCOL) v3 (Stenke et al. (2013); Revell et al. (2015)) | T42 | 39 (0.01 hPa) | Free-Running | Nordeng (1994) |
| NIES-C1SD | CCSRNIES-MIROC3.2 (Imai et al. (2013); Akiyoshi et al. (2016)) | T42 | 34 (0.01 hPa) | ERA-Interim (Nudging) | Arakawa and Schubert (1974) |
| NIES-C1 | " " | " " | " " | Free-Running | " " |
| MOCAGE-CTM | Modele de Chimie Atmosphérique de Grande Echelle (MOCAGE) (Josse et al. (2004); Guth et al. (2016)) | 2° x 2° | 47 (5 hPa) | ERA-Interim (CTM) | Bechtold et al. (2001) |
| ULAQ-C1 | University of L'Aquila (ULAQ)-CCM (Pitari et al. (2014)) | T21 | 126 (0.04 hPa) | Free-Running | Grewe et al. (2001) |
| ACCESS-C1 | National Institute of Water and Atmospheric Research UK Chemistry and Aerosols (NIWA-UKCA) (Morgenstern et al. (2009, 2013); Stone et al. (2016)) | 3.75° x 2.5° | 60 (84 km) | Free-Running | Hewitt et al. (2011) |

**Table 1** Details of the model integrations, where columns 3-6 correspond to the horizontal resolution, number of vertical levels and model top, source of meteorological fields and reference for the model's convective parameterizations. T21 and T42 correspond to quadratic grids of approximately to $\sim 5.6° \times 5.6°$ and $\sim 2.8° \times 2.8°$, respectively. Two types of model experiments are examined: the REF-C1SD and REF-C1 simulations. Both REF-C1 and REF-C1SD are constrained with observed sea surface temperatures (SSTs) and sea ice concentrations (SICs). Whereas the REF-C1 experiment use internally generated (or free-running) meteorological fields, however, the C1SD or C1 "Specified Dynamics" simulations is constrained with (re)analysis meteorological fields. Note that both CTMs and nudged simulations are included in the REF-C1SD suite of model simulations (see column 6). In cases where individual modeling agencies performed multiple C1SD simulations (e.g. NASA and NCAR) we append a "V1/V2" to the simulation name.



# Idealized Tracers

| Tracer ($\chi$) | Boundary Condition ($\chi$ ) | Source (S) |
|---|---|---|
| 5-Day NH-Loss ($\chi_5$) | 1 over $\Omega_{\mathrm{MID}}$ | $-\chi \tau$ ($\tau$ = 5 days, entire atmosphere) |
| 50-Day NH-Loss ($\chi_{50}$) | 1 over $\Omega_{\mathrm{MID}}$ | $-\chi \tau$ ($\tau$ = 50 days, entire atmosphere) |
| Tropospheric Mean Age ($\Gamma_{\mathrm{NH}}$) | 0 over $\Omega_{\mathrm{MID}}$ | 1 year/year |
| Stratospheric-Loss ($\chi_{\mathrm{STE}}$) | 200 ppbv above 80 hPa | $-\chi \tau$ ($\tau$ = 25 days, troposphere only) |
| Global Source Decay (e90) | 100 ppbv in first model level | $-\chi \tau$ ($\tau$ = 90 days, entire atmosphere) |

**Table 2** Table of idealized tracers, $\chi$, integrated in the simulations. All tracers ($\chi$) satisfy the tracer continuity equation, $(\partial_t + \mathcal{T})\chi(\mathbf{r}, t|)$ $= S$ in the interior of the atmosphere (that is, outside of ), where $\mathcal{T}$ is the linear advection-diffusion transport operator and $S$ denotes interior sources and sinks. For the first three tracers (rows 2-4) is taken to be the NH midlatitude surface, $_{\mathrm{MID}}$, which is defined throughout as the first model level spanning latitudes between 30°N and 50°N. The last two tracers, referred to throughout as the global source tracers, include the stratospheric tracer $\chi_{\mathrm{STE}}$, which is set to 200 ppbv for pressures less than and equal to 80 hPa, and decays uniformly in the troposphere at a loss rate $\tau_d = 25$ days $^{-1}$ (row 5). The e90 tracer is uniformly emitted over the entire surface layer and decays exponentially at a rate of 90 days $^{-1}$ such that concentrations greater than 125 ppb tend to reside in the lower troposphere and concentrations less than 50 ppb reside in the stratosphere (row 6).



# Model Output

| Simulation Name | $\chi_5$ | $\chi_{50}$ | $\Gamma_{NH}$ | $\chi_{STE}$ | e90 | $U$ | $V$ | $T$ | $\omega$ | CMF |
|---|---|---|---|---|---|---|---|---|---|---|
| GEOS-CTM | X | X | X | X | X | X | X | X | X | X |
| GEOS-C1SD | X | X | X | X | X | X | X | X | X | X |
| GEOS-C1 | X | X | X | X | X | X | X | X | X | X |
| WACCM-C1SDV1 | X | X | X | X | X | X | X | X | X | X |
| WACCM-C1SDV2 | X | X | X | X | X | X | X | X | X | X |
| WACCM-C1 | X | X | X | X | X | X | X | X | X | X |
| **CAM-C1SD** | X | X | X | X | X | X | X | X | X | X |
| CAM-C1 | X | X | X | X | X | X | X | X | X | X |
| EMAC-L47-C1 | X | X | | * | X | X | X | X | X | X |
| EMAC-L47-C1SD | X | X | | * | X | X | X | X | X | X |
| EMAC-L90-C1 | X | X | | * | X | X | X | X | X | X |
| EMAC-L90-C1SD | X | X | | * | X | X | X | X | X | X |
| MRI-C1SD | X | X | X | | | X | X | X | X | X |
| MRI-C1 | X | X | X | | | X | X | X | X | X |
| CMAM-C1SD | | | X | X | X | X | X | X | X | X |
| CMAM-C1 | | | X | X | X | X | X | X | X | X |
| NIWA-C1 | X | X | | X | | X | X | X | X | X |
| SOCOL-C1 | X | X | | * | | X | X | X | X | * |
| NIES-C1SD | X | X | * | * | | X | X | X | X | |
| NIES-C1 | X | X | * | * | | X | X | X | X | |
| MOCAGE-CTM | X | X | X | | * | X | | X | | |
| ULAQ-C1 | X | X | X | X | X | X | X | X | X | |
| ACCESS-C1 | X | X | | X | | X | X | X | X | X |

*incorrectly implemented

**Table 3** List of the model simulations for which the idealized tracers ($\chi_5$, $\chi_{50}$, $\Gamma_{NH}$, $\chi_{STE}$ and e90) and dynamical fields (U, V, $\omega$, T and parameterized convective mass fluxes (CMF)) were available. Asterisks denote fields that were output in simulations, but were not correctly implemented.