# Peer review of "Large-Scale Tropospheric Transport in the Chemistry Climate Model Initiative (CCMI) Simulations"

_Atmospheric Chemistry and Physics, 2017_

## Referee Comment (RC1) · Anonymous Referee #1 · 30 Dec 2017

The paper shows an intercomparison of tropospheric transport among several models participating in the CCMI project, using a suite of artificial tracers. The results reveal substantial inconsistencies in transport which can be attributed mainly to convective parameterizations. This is suggested by the qualitative comparison of the spread in tracer concentrations compared to the spread in large-scale winds, and supported by the fact that nudging the dynamical fields to reanalysis data the does not improve the comparison. The paper is well written and easy to follow, and provides an insightful overview of the degree of agreement on general tropospheric transport properties among state-of-the-art models. The following minor revisions are suggested before publication in ACP, which are points in the text that need clarification and figures that

need some improvement.

- P6 L15: the global tracers are also 'idealized loss tracers'

- Fig. 4 : the legend is missing the purple points (ULAQ?)

- In all figures, several members are considered for some simulations. Please mention this somewhere. Also, there seem to be several members of the specified dynamics runs, what is the point of this if the dynamical fields are nudged and why do they differ substantially (e.g. Fig. 4b)?

- P6 L16-19: It is not clear to me why an inverse relationship is expected between the global and the midlatitude tracers, if all are emitted at the surface and subject to the same convective and isentropic transport? Could you explain why does the dilution argument only apply to the midlatitude tracers?

- P6 L16-19: I don't see the blue and red curves being particularly low in Fig. 3a-b. This is true only for the comparison of these two curves with ULAQ. Could you clarify what you mean? Are you referring to the 30-50N band?

- Fig. 2 and 3- I suggest revising the legend to match the lines shown in the figures. What model does the orange solid line refer to? And the thin brown line? Figures 1 and 2: why are there solid lines in the REFC1SD panels corresponding to the EMAC model? Should these be dashed? Fig. 5: it is hard to distinguish the multi-model mean from the EMAC lines.

- P6 L31: That paper uses future runs, which cannot be Specified dynamics.

- P6 L33-34, Supplementary Figure 2 and Table 3: It would be helpful to briefly explain what exactly was (wrongly) implemented in the STE tracer for each of the runs.

P7 L 2-6: It would be easier for the reader if you pointed to specific longitudes when you refer to regions such as 'over the oceans', 'downstream of the storm tracks' or 'over land'.

- Fig. 6c: The midlatitude convection box located over south-west Asia is not really capturing midlatitude convection, and there is not much convection over most of the box. Instead this box could be placed over central Europe, where there is significant summertime convection.

- Figs. 6 and 7: What are the units of CMF?

- P8 L2-4: Is this true also for the other tracers (X50 and e90)?

- P8 L16-18: Although a useful comparison of the large-scale flow, Supplementary Fig. 3 does not inform on the 'relationship between large-scale flow biases over NH midlatitudes and the transport differences among the simulations'. Could you rephrase or add information to justify the claim?

- P9 L12: Could you give an approximate % value of the bias?

- Fig. 11 caption: remove 'strong'

- Table 2: Its seems that some symbols have disappeared, please revise.

---

## Referee Comment (RC2) · Anonymous Referee #2 · 4 Jan 2018

This study examined the large-scale tropospheric transport in the chemistry-climate model initiative (CCMI) ensemble. As to the transport from the Northern Hemisphere midlatitude surface into the Arctic, the inter-model spreads reflect differences in parameterized convection over the northern midlatitude oceans, especially in boreal winter. In contrast, the inter-model differences in interhemispheric transport from northern midlatitudes to southern high latitudes are mostly due to differences in tropical and subtropical convection. It is also found that the simulations constrained by analyzed winds show at least as large differences as the simulations using internally generated meteorological fields, owing to differences in parameterized convection. The results are very interesting and clearly presented. My comments are very minor.

[Figure]

P4, L22: When the data are interpolated in pressure, were the pressure levels below the ground treated by missing values or linear interpolation?

P4, L30: This schematic refers to the mean meridional circulation only in the tropics

P7, L17-19: It might be noted that the tracer isolines tend to be more vertical than the isentropic surfaces in summer compared to winter, which indicates moisture may also contribute to the isolation of the Arctic from midlatitudes in summer — latent heat release allows moist air parcel rising along a front faster than a dry parcel.

General comment: Since these model transport differences can be attributed to differences in parameterized convection, it would be hard to reduce the transport uncertainties because we don't know which convective schemes are better. It seems that future efforts should be focused on comparing idealized tracers with realistic tracers that are available in observations, which may help to reduce the uncertainty in convective schemes.

---

## Author Comment (AC1) · 8 Mar 2018

Manuscript number: acp-2017-1038 Title: "Large-Scale Tropospheric Transport in the Chemistry Climate Model Initiative (CCMI) Simulations"

Dear Dr. Peter Hess:

We thank the referees for their reviews of our manuscript. The generally positive tone of the reviews is encouraging. After careful consideration of all reviewers' suggestions we feel that the changes that we have made to the text have improved the manuscript. Before responding to the referees point-by-point, we first address the main issues that

the referees raised.

We very much appreciate Reviewer 1's critical reading of the manuscript. In particular, he/she highlighted that the inverse relationship between the global surface (e90) and Northern Hemisphere midlatitude loss tracers shown in Figure 4b is not intuitive, given that both tracers are subject to prescribed mixing ratios at the surface. We have since clarified in the text that this relationship depends sensitively on latitude and ultimately reflects differences in the meridional gradients of the tracers, consistent with differences in their surface boundary conditions. We feel that this section has improved as a result of the referee's close reading of the manuscript, although our analysis still remains somewhat limited by the fact that we lack high temporal tracer output that we would need to address the referee's concerns more thoroughly (i.e. to construct closed tracer budgets). We hope that our changes are satisfactory to the referee. We also appreciate her/his comments regarding several plotting errors in our figures and ambiguity in some passages in the text.

We also agree with Reviewer 2's general comment that we should more directly address the possibility of constraining tropospheric transport from real observable tracers. To this end we have added a paragraph in the Conclusions section that addresses ways in which combinations of trace gases, including chlorofluorocarbons and sulfur hexafluoride, may be used to infer different aspects of the transit-time distribution (TTD) from observations. Given that previous studies show that the idealized loss tracers may be used to approximate some aspects of this distribution, future research efforts should be paid to extracting more observational estimates of the TTD. We hope that this paragraph addresses the referee's concerns.

We have considered all comments carefully and modified our manuscript accordingly. We have provided two versions of the revised manuscript, one of which includes the corrections highlighted in red. The point-by-point responses to the referee's comments are also attached. We hope that the manuscript is now acceptable for publication in ACP. I confirm that my coauthors, Huang Yang, Darryn W. Waugh, Guang Zeng, Olaf

Morgenstern, Douglas E. Kinnison, Jean-Francois Lamarque, Simone Tilmes, David A. Plummer, John F. Scinocca, Beatrice Josse, Virginie Marecal, Patrick J\"ockel, Luke D. Oman, Susan E. Strahan, Makoto Deushi, Taichu Y. Tanaka, Kohei Yoshida, Hideharu Akiyoshi, Yousuke Yamashita, Andreas Stenke, Laura Revell, Timofei Sukhodolov, Eugene Rozanov, Giovanni Pitari, Daniele Visioni, Kane A. Stone, Robyn Schofield and Antara Banerjee concur with the submission of our manuscript in its revised form. The revised version of the manuscript has been resubmitted electronically.

Sincerely, Clara Orbe
* * *
Response to Reviewer 1:

We thank the reviewer for his/her insightful comments and close reading of the manuscript. Please see the attached manuscript with changes highlighted in red. Our responses are as follows:

Response to Minor Comments:

Comment #1: P6 L15: the global tracers are also 'idealized loss tracers'.

Good point. We have corrected this in the text.

Comment #2: Fig. 4: the legend is missing the purple points (ULAQ?)

Thank you for catching this oversight! We have now added those points to the legend.

Comment #3: In all figures, several members are considered for some simulations. Please mention this somewhere. Also, there seem to be several members of the specified dynamics runs, what is the point of this if the dynamical fields are nudged and why do they differ substantially (e.g. Fig. 4b)?

We only use one ensemble member per modeling group, despite the fact that several modeling groups submitted more than one ensemble member. We now state this

clearly in the Methods section, where we refer to using only the first ("r1i1p1") ensemble member for the REF-C1 and REF-C1SD experiments. With regards to the specified dynamics simulations WACCM-C1SDV1 and WACCM-C1SDV2, we hope to clarify to the referee that these are not ensemble members of the same experiment, but rather distinct simulations that use two different relaxation (nudging) times. Furthermore, one of the main conclusions from this study is that, even among simulations that use the same reanalysis fields, we find that there are large differences in their global-scale transport properties related to large differences in (parameterized) convection. Indeed, in some cases, these differences are larger than the convection differences among FR simulations using the same models. We hope that we are being clear.

Comment #4: P6 L16-L19: Could you explain why does the dilution argument only apply to the midlatitude tracers.

Thank you for the comment. We agree with the referee that this is not obvious and more care should have been taken in handling this argument. Indeed, the relationship between the midlatitude tracers and e90 depends sensitively on latitude, a point that we failed to mention in the text. Ultimately, this reflects different (at places, opposite) meridional gradients in the tracer, which are related to the fact that the 5-day and 50-day loss tracers are subject to prescribed mixing ratios over the NH midlatitude surface, whereas e90 is prescribed globally at the surface. We have amended the text as follows:

"Over the middle and northern edge of the midlatitude source region, however, the tracers exhibit an inverse (and relatively compact) relationship (Figure 4b). While this inverse relationship is not intuitive, it is consistent with differences in the meridional gradients of the tracers, wherein $X_5$ (e90) increases (decreases) moving poleward from the northern subtropics over northern midlatitudes. Perhaps fortuitously, the NH midlatitude tracers are only sourced in the region of strongest isentropic mixing so that $X_5$ always decreases along an isentropic surface as ones moves from the midlatitude surface poleward to the Arctic (Fig. 1a). By comparison, e90 features its largest

concentrations over the Arctic (Fig. 1d) so that stronger mixing over midlatitudes can actually dilute tracer mixing ratios along a given isentrope. Thus, the relationship between the surface sourced tracers is not straightforward, but rather sensitive to how two-way mixing operates on different (and, at places, opposite) along-isentropic tracer gradients. More work is needed to disentangle this relationship but is beyond the scope of the current study."

While the lack of high temporal model output (needed for calculating tracer budgets) precludes a further investigation of the relative roles of advection versus mixing on the tracer concentrations in this region we expect to pursue this further in future studies. We hope that the reviewer understands that we are limited by the available output. We will wait to hear back if our response is satisfactory.

Comment #5:P6: I don't see the blue and red curves being particularly low in Fig. 3a-b. This is true only for the comparison of these curves with ULAQ. Could you clarify what you mean? Are you referring to the 30-50N band?

Thank you for the comment. We agree with the referee that we were not being clear. Yes, we are referring to southern edge of the NH midlatitude (30-50N) band, over which the red and blue dashed lines (Fig 3b) are low not only with respect to the ULAQ model, but also compared to the free-running simulations using the same models (Fig 3a). We have clarified this now in the text. Please also see our response to the previous comment.

Comment #6: Fig. 2 and 3- I suggest revising the legend to match the lines shown in the figures. What model does the orange solid line refer to? And the thin brown line? Figures 1 and 2: why are there solid lines in the REFC1SD panels corresponding to the EMAC model? Should these be dashed? Fig. 5: it is hard to distinguish the multi-model mean from the EMAC lines.

As before, many thanks for this comment. Indeed, we did not include the orange (CMAM) lines in the legend and this has been fixed in the current version of the

manuscript. Our apologies for any confusion this has caused. Regarding the solid lines in the REFC1SD panels in Figure 2, we hope to clarify that the grey lines in each panel always refer to the simulations that are not either REF-C1 (in left panels in) or REF-C1SD (in right panels). We never use dashed grey lines. This is mentioned in the caption to that figure and will wait to hear back from the referee is she/he requests further changes. Finally, regarding Figure 5 we understand that the multi-model mean (grey) line may be hard to distinguish from the EMAC lines. For that reason we have changed to a darker (and thicker) grey line. We have changed this in the new versions of Figures 5 and 7 (for consistency).

Comment #7: P6 L31: That paper uses future runs, which cannot be Specified dynamics

Thank you for pointing out this mistake! We have corrected this in the text. Please see the red comments in the revised manuscript.

Comment #8: P6 L33-34, Supplementary Figure 2 and Table 3: It would be helpful to briefly explain what exactly was (wrongly) implemented in the STE tracer for each of the runs.

Thanks for the comment. We have added a sentence at the end of that paragraph emphasizing that care must be taken when analyzing the STE tracer output as several modeling groups applied the tracer's chemical loss incorrectly (i.e. below 80 hPa, not the tropopause).

Comment #9: P7 L 2-6: It would be easier for the reader if you pointed to specific longitudes when you refer to regions such as 'over the oceans', 'downstream of the storm tracks' or 'over land'.

We agree with the referee that adding longitude references would help orient the reader. Please see the new text on Page 7 where we have added those changes.

Comment #10: Fig. 6c: The midlatitude convection box located over south-west Asia

is not really capturing midlatitude convection, and there is not much convection over most of the box. Instead this box could be placed over central Europe, where there is significant summertime convection.

We agree with the referee that there are much larger values of CMF over Central Europe. However, note that this convection is north of the midlatitude origin region (as defined in this study) which spans latitudes between 30N and 50N. Furthermore, the mid-tropospheric isentropic surfaces intersect midlatitudes over South/Central Asia, not over Europe, so that this box spans the region that is most important for lifting boundary layer air aloft into the middle and upper troposphere. For both reasons, therefore, we keep the box centered over Asia.

Comment #11: Figs. 6 and 7: What are the units of CMF?

The units of CMF are kg/m2/s (mass flux). This is already noted on the colorbars in Figure 6 and on the horizontal axes in Figure 7. We will wait to hear back if the referee prefers that we place these labels elsewhere. Otherwise, the figures have not been changed.

Comment #12: P8 L2-4: Is this true also for the other tracers (X50 and e90)?

This is also true for the longer-lived (and global source) tropospheric loss tracers, as they feature similar gradients over northern midlatitudes. Because their vertical gradients are weaker, however, the impact of convection on the strength of the vertical profile is slightly weaker.

Comment #13: P8 L16-18: Although a useful comparison of the large-scale flow, Supplementary Fig. 3 does not inform on the 'relationship' between large-scale flow biases over NH midlatitudes and the transport differences among the simulations? Could you rephrase or add information to justify the claim?

Thank you for pointing this. Indeed, we are being generous in that claim. We simply mean to comment that there is no obvious relationship between the large-scale vertical

velocity strength and the tracer ages among the CCMI simulations. We have added this caveat to the manuscript as follows:

"Note that a more rigorous examination comparing the large-scale flow and transport biases among the simulations is not presented here and would be more appropriate using sub-monthly output (for constructing tracer budgets). As such, our inference here is qualitative."

Indeed, had we been provided with daily output from the simulations we would hope to have been able to do tracer budget decompositions in terms of the residual mean circulation. One example of such an analysis is presented in the following study (for the e90 tracer and for one model only):

Abalos, Marta, William J. Randel, Douglas E. Kinnison, and Rolando R. Garcia. "Using the artificial tracer e90 to examine present and future UTLS tracer transport in WACCM." Journal of the Atmospheric Sciences, 74, no. 10 (2017): 3383-3403.

Comment #14: P9 L12: Could you give an approximate % value of the bias?

Yes – this is a good suggestion. We have now added the following clause to the end of that line: "by 20-40% for most of the models but up to 60% for others."

Comment #15: Fig. 11 caption: remove 'strong'

Thanks – we agree. We have removed that word from the caption.

Comment #16: Table 2: It seems that some symbols have disappeared, please revise.

Thank you for catching this! Sorry that these do not appear clearly. We have changed this accordingly. Please see the new table.

———————————————————————————————————

Response to Reviewer 2:

We thank the reviewer for his/her comments and careful reading of the manuscript.

Please see the attached manuscript with changes highlighted in red. Our responses are as follows.

Response to Major Comment:

Comment #1: Since these model transport differences can be attributed to differences in parameterized convection, it would be hard to reduce the transport uncertainties because we don't know which convective schemes are better. It seems that future efforts should be focused on comparing idealized tracers with realistic tracers that are available in observations, which may help to reduce the uncertainty in convective schemes.

We agree with the referee that observational constraints of the idealized tracers are needed to discern which models are "better." The mean age with respect to the NH midlatitude surface can be compared with surface observations of sulfur hexafluoride, after being recast in terms of an "SF6-age" as in Waugh et al. 2013. The estimates from that study indicate that the mean ages in the CCMI simulations are old, compared to the observations. However, more work is needed to identify observable species that can be used to constrain other aspects of the underlying transit-time distribution (of which the mean is the "mean age"). One approach that looks promising is presented in another study (Holzer and Waugh (2015), now cited), in which the authors use combinations of different chlorofluorocarbons (CFCs) and CFC-replacement compounds to constrain the TTD connecting the SH to the NH midlatitude surface. More work is needed to extend this approach to tracers with different source regions and to different regions in the troposphere. We now mention these issues in a new paragraph that we have added to the Conclusions section. We hope that this addresses the reviewer's concerns.

Response to Minor Comments:

Comment #1: P4, L22: When the data are interpolated in pressure, were the pressure levels below the ground treated by missing values or linear interpolation?

Our apologies for not mentioning this in the text. We treat values below the ground

[Figure]

as missing (NaN) values. We have now added a sentence to the methods section mentioning this. Thanks for the comment.

Comment #2: P4, L30: This schematic refers to the mean meridional circulation only in the tropics

Thank you for catching this! Indeed, in a previous version of the figure we had overlaid a schematic of the residual mean (thermally direct) circulation as it extends out to the extratropics. However, we failed to correct the corresponding text, which we have now revised. Thank you for the comment.

Comment #3: It might be noted that the tracer isolines tend to be more vertical than the isentropic surfaces in summer compared to winter, which indicates moisture may also contribute to the isolation of the Arctic from midlatitudes in summer latent heat release allows moist air parcel rising along a front faster than a dry parcel.

We agree with the referee that this is an important point to include in the text. Indeed, because of transport associated with moist latent heat release the tracer isolines are not parallel to the (dry) surfaces of constant potential temperature (neither in summer nor winter). We have included this caveat in the text. Thanks for the comment.

Please also note the supplement to this comment:
https://www.atmos-chem-phys-discuss.net/acp-2017-1038/acp-2017-1038-AC1-supplement.pdf

**Supplement:**

[revised manuscript text omitted]